**Brief Communication**

# PTM-Mamba: a PTM-aware protein language model with bidirectional gated Mamba blocks

Fred Zhangzhi Peng[1], Chentong Wang[2], Tong Chen[1], Benjamin Schussheim[3], Sophia Vincoff[1] & Pranam Chatterjee [1,3,4] ✉

Current protein language models (LMs) accurately encode protein properties but have yet to represent post-translational modifications (PTMs), which are crucial for proteomic diversity and influence protein structure, function and interactions. To address this gap, we develop PTM-Mamba, a PTM-aware protein LM that integrates PTM tokens using bidirectional Mamba blocks fused with ESM-2 protein LM embeddings via a newly developed gating mechanism. PTM-Mamba uniquely models both wild-type and PTM sequences, enabling downstream tasks such as disease association and druggability prediction, PTM effect prediction on protein–protein interactions and zero-shot PTM discovery. In total, our work establishes PTM-Mamba as a foundational tool for PTM-aware protein modeling and design.

PTMs, such as phosphorylation, acetylation, ubiquitination and glycosylation, vastly expand the functional diversity of eukaryotic proteomes, influencing essential processes like enzyme activity, protein turnover, signaling cascades and DNA repair[1,2]. Dysregulation of PTMs often leads to severe diseases, including cancer, neurodegeneration and aging[2,3]. For example, phosphorylation of STAT3 at specific residues transforms it from a typical transcription factor into a driver of tumorigenesis and metastasis in various cancers[4,5]. Understanding and modeling the unique sequence features of post-translationally modified proteins is therefore crucial for advancing proteome-wide insights and therapeutic design. Protein LMs have emerged as transformative tools for encoding physicochemical and functional information in protein sequences[6]. Models like ESM-2 and ProtT5 excel at sequence representation, whereas autoregressive protein LMs like ProGen and ProtGPT2 generate functional proteins[7–10]. From a therapeutic context, our generative language models, such as SaLT&PepPr, PepPrCLIP, PepMLM and moPPIt, have enabled the design of peptides that bind and degrade specific targets, including disordered proteins[11–14]. However, existing protein LMs entirely exclude PTM residues from their training and inference pipelines[7–10], limiting their ability to model PTM-specific effects.

We hypothesized that combining ESM-2 embeddings with a specialized framework for handling PTM tokens would enable accurate modeling of both wild-type residues and PTMs. To test this, we curated a training dataset of 79,707 modified sequences, constructed from 311,350 experimentally validated PTM records in Swiss-Prot[15]. We specifically mapped PTM annotations to their respective protein sequences, ensuring a diverse representation of PTM types (Supplementary Fig. 1) and sequence lengths (Supplementary Fig. 2).

We based our PTM protein LM on Mamba, a structured state-space model that offers computational efficiency and flexibility through a selective state-space architecture, which provides subquadratic time and memory complexity with sequence length[16]. Additionally, Mamba uses hardware-aware primitives, such as parallelized state transitions and convolutional projections, to accelerate computations without affecting scaling[16]. Although Mamba's original design for autoregressive text generation limited its ability to capture full sequence semantics, we adapted it for bidirectional modeling by introducing forward and backward processing layers. The resulting bidirectional Mamba block (Fig. 1a and code snippet below) processes the sequence in two directions: a forward pass (left to right) and a backward pass (right to left). Each pass independently generates hidden states through its respective state-space layer, and the outputs are concatenated before being fused by a fully connected layer to generate a combined representation. Residual connections are applied to both the forward and backward layers, and their contributions are averaged to retain both

[1]Department of Biomedical Engineering, Duke University, Durham, NC, USA. [2]School of Life Sciences, Westlake University, Hangzhou, China. [3]Department of Computer Science, Duke University, Durham, NC, USA. [4]Department of Biostatistics and Bioinformatics, Duke University, Durham, NC, USA. ✉e-mail: pranam.chatterjee@duke.edu

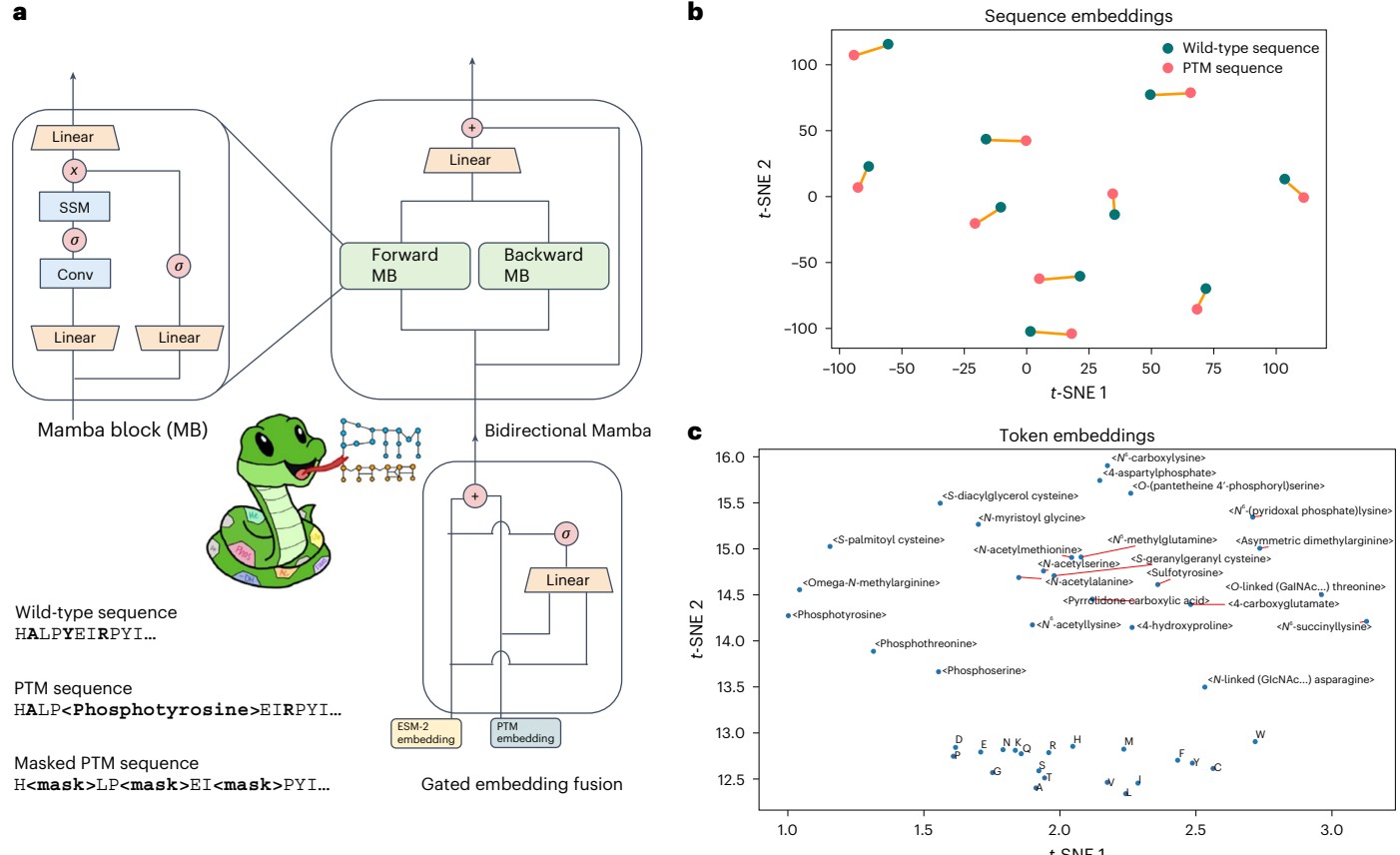

**Fig. 1 | Architecture and embedding visualization of PTM-Mamba. a**, Primitives of PTM-Mamba. Bottom left, given a sequence, with 80% probability, we perform standard 15% token masking, and, with 20% probability, we mask all the PTM tokens and randomly mask 15% of wild-type tokens. The bidirectional Mamba block in PTM-Mamba is built on top of the Mamba block (MB), which processes the sequences in both the forward (forward Mamba block) and backward (backward Mamba block) orientation. The gated embedding fusion module inputs ESM-2 and PTM embeddings and fuses them in a gated manner via a sigmoid-activated linear layer. SSM, state-space model. **b**, *t*-SNE visualization of PTM-Mamba embeddings of select wild-type and corresponding PTM protein sequences. Orange lines connect the corresponding embeddings. **c**, *t*-SNE visualization of labeled token embeddings. Conv, local 1D convolutional layer.

directional contexts, ensuring comprehensive modeling of sequence dependencies for amino acids and PTMs.

```
def bidirectional_mamba(self, hidden_states):
  residual = None
  for f_layer, b_layer, h_fc in zip(
      self.forward_layers, self.backward_layers,
self.hidden_fc
    ):
    hidden_states_f, residual_f = f_layer(
      hidden_states, residual,
    )
    flip_residual = residual.flip([1]) if residual
is not None else None
    hidden_states_b, residual_b = b_layer(
      hidden_states.flip([1]), flip_residual,
    )
    hidden_states = h_fc(
      torch.cat([hidden_states_f, hidden_states_
b.flip([1])], dim=-1)
    )
    residual = 0.5 * (residual_f + residual_
b.flip([1]))
```

To preserve comprehension of regular amino acids, we trained our new PTM-Mamba model as a head to the state-of-the-art ESM-2-650M

model[7], in which wild-type amino acid tokens are passed into ESM-2-650M to retrieve its output embeddings and PTM tokens are converted into <mask> tokens for ESM-2-650M input (Fig. 1a). Sequences are finally fed into the embedding layer of PTM-Mamba, which naturally processes both wild-type and PTM tokens. To join the ESM-2-650M and PTM-Mamba embeddings, we propose a new gating mechanism in which the two embeddings are concatenated and filtered via a sigmoid-activated linear gate to produce a final output representation (Fig. 1a and code snippet below).

```
def gated_fuse(input_ids, esm_embedding):
  ptm_mamba_embedding = Embedding(input_ids)
  gate = Linear(torch.cat([hidden_states,
esm_embedding], dim=-1)).sigmoid()
  hidden_states = ptm_mamba_embedding * gate +
esm_embedding * (1 - gate)
  return hidden_states
```

We compared PTM-Mamba to a baseline PTM-Transformer model and observed faster convergence on training accuracy (Supplementary Fig. 3), highlighting the comparative efficiency of the bidirectional Mamba blocks and gating mechanism. Beyond efficiency, the primary objective of PTM-Mamba is to distinctly, yet relevantly, represent both unmodified and post-translationally modified sequences, capturing the critical biological functions and structural changes induced by PTMs. To assess this capability, we visualized

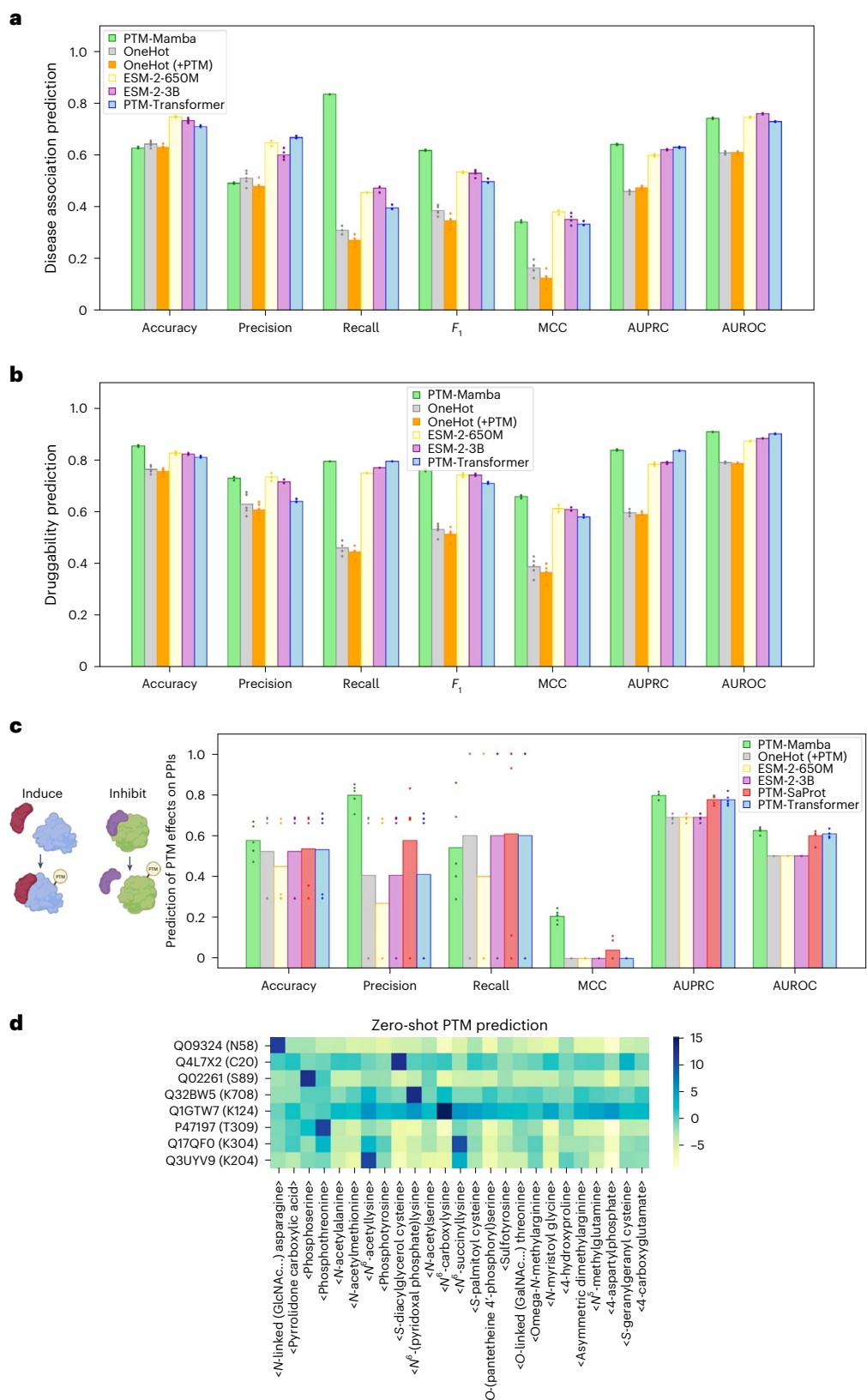

**Fig. 2 | Performance evaluation of PTM-Mamba across diverse PTM-related tasks. a**, Disease association prediction for PTM modified sequences, evaluated across accuracy, precision, recall, $F_1$, MCC, area under the precision–recall curve (AUPRC) and area under the receiver operating characteristic curve (AUROC). **b**, Druggability prediction of PTM modified sequences, evaluated across the same metrics. **c**, Prediction of PTM effects on PPIs, using PTMint data to classify whether a PTM induces or inhibits an interaction. All benchmarks in **a**–**c** were performed with replicates ($n = 5$). **d**, Visualization of the predicted logits for zero-shot PTM discovery. Rows denote different amino acids in the format of 'Uniref-accession-id (amino-acid position)', and columns denote the logit value of the PTMs. Schematic in panel **c** created using BioRender.com.

PTM-Mamba embeddings using *t*-distributed stochastic neighbor embedding (*t*-SNE). The embeddings revealed a nuanced distinction between wild-type protein sequences and their PTM modified counterparts, with embeddings for each wild-type pair in close proximity (Fig. 1b). This suggests the ability of PTM-Mamba to capture the subtle yet notable effects of PTMs while maintaining the contextual integrity of the protein sequence. Additionally, token embeddings for PTM residues showed class-specific organization, with spatial proximity observed among tokens for phosphorylation and acetylation as examples (Fig. 1c). PTM residue tokens also exhibited greater spatial diversity than wild-type tokens, reflecting the model's focus on encoding PTM-specific information (Fig. 1c).

To confirm that PTM-Mamba embeddings maintain strong performance on standard PTM prediction tasks, we evaluated them on phosphorylation site prediction (Supplementary Fig. 4) and non-histone acetylation site prediction (Supplementary Fig. 5). Using curated datasets for both tasks, we conducted per-residue binary classification and compared PTM-Mamba embeddings against baselines, including ESM-2-650M, ESM-2-3B, PTM-Transformer and baseline one-hot embeddings. PTM-Mamba maintained comparable performance across all metrics, confirming that its embeddings retain general applicability for PTM-related tasks. Notably, these tasks do not explicitly represent PTM tokens, which aligns with the observation that PTM-Mamba is primarily optimized for use cases involving modified sequences, rather than wild-type-only benchmarks.

We next evaluated PTM-Mamba on three benchmarking tasks explicitly leveraging PTM tokenization: disease association prediction, druggability prediction and the effects of PTMs on protein–protein interactions (PPIs). For disease association prediction, we used a dataset curated from the dbPTM database[17] that links PTMs to conditions such as cancer, neurodegenerative disorders and diabetes, with annotations sourced from databases such as PhosphoSitePlus, ActiveDriverDB and genome-wide association studies (GWAS) as well as manual curation[18,19]. Druggability prediction assessed PTM sequences that influence therapeutic targetability, focusing on how modifications alter protein structure and accessibility of binding sites[17]. To evaluate the effects of PTMs on PPIs, we used the PTMint dataset, which annotates experimentally validated PTM-mediated regulatory roles, specifically whether a PTM induces or inhibits a PPI[20]. For all tasks, wild-type sequences were mapped to PTM-Mamba's dataset, with residues replaced by the corresponding PTMs for tokenization, while baseline models, including one-hot embeddings and ESM-2 embeddings, used wild-type sequences as input.

For disease association prediction, PTM-Mamba performs strongly versus baseline models, including ESM-2-650M and PTM-Transformer, demonstrating its ability to capture PTM-specific effects essential for identifying disease-associated proteins (Fig. 2a). Similarly, for druggability prediction, PTM-Mamba achieved robust performance, often exceeding baselines across key metrics such as $F_1$ score and Matthews correlation coefficient (MCC), highlighting its relevance for therapeutic design (Fig. 2b). For the key PTM effect on the PPI task, PTM-Mamba achieved the highest metrics among all models, including PTM-Transformer and PTM-SaProt, a novel baseline model that replaces ESM-2 with state-of-the-art, structure-aware SaProt protein LM embeddings[21], indicating that sequence-focused models may capture PTM effects more optimally (Fig. 2c). This benchmark showcases PTM-Mamba's ability to model complex regulatory dynamics mediated by PTMs, further highlighting its utility for biologically relevant downstream applications.

Finally, we explored PTM-Mamba's utility for zero-shot PTM discovery, a task of great biological relevance. By analyzing model logits for masked positions in wild-type sequences, PTM-Mamba accurately predicted plausible PTMs for specific residues, such as <phosphoserine> for serine in UniProt sequence Q02261 and <*S*-diacylglycerol cysteine> for cysteine in UniProt sequence Q4L7X2 (Fig. 2d). This

capability offers PTM-Mamba as a tool for biologists to generate new insights into PTM biology without requiring additional training or labels.

In total, PTM-Mamba provides new opportunities for modeling and designing PTM-specific protein sequences, particularly via its ability to explicitly tokenize PTM modified proteoforms for applications ranging from disease mechanism studies to therapeutic design with enhanced targeting specificity. For future work, we plan to address the limited availability of experimentally validated PTM annotations by augmenting the training dataset using mass spectrometry-based PTM databases[22]. We also aim to explore structure prediction of PTM modified sequences as a new task that can leverage PTM-Mamba's embeddings, alongside extending these embeddings to design PTM-specific binders that selectively target modified protein states[6,23,24]. Together, by enabling PTM-aware modeling, PTM-Mamba has the potential to reshape proteome analysis and drive innovation in precision therapeutics.

## Online content

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

## Methods

### Data curation

Model training data were curated from UniProt[15]. Specifically, 311,350 experimentally validated PTM records were collected from Swiss-Prot, and the PTM annotations of their proteins were mapped to their respective sequences to construct the new PTM sequences. The final dataset includes a total of 79,707 PTM sequences. Data curation code can be found at https://github.com/programmablebio/ptm-mamba/tree/main/ptm_data_preprocessing.

Datasets for the four benchmarking tasks were collected from the following sources. Phosphorylation site data were obtained from the corresponding ProteinBERT benchmark[25], originally derived from PhospoSitePlus[18] and filtered for sequences between 256 and 512 amino acids in length, yielding a training set of 15,588 sequences, a validation set of 1,707 sequences and a testing set of 3,106 sequences. Non-histone acetylation site prediction was performed equivalently as described in prior literature, using the non-histone acetylation collection dataset[26]. Druggability and disease association datasets were curated from the dbPTM database[17]. PPI data describing the effect of PTMs were curated from PTMint, which encompasses 2,477 nonredundant PTM sites in 1,169 proteins affecting 2,371 protein–protein pairs[20]. In brief, wild-type sequences were mapped to corresponding entries in the PTM-Mamba dataset, and wild-type residues were replaced by the corresponding position-specific PTMs for tokenization by specified models. For all other baseline models trained with standard one-hot embeddings or ESM-2 embeddings, the corresponding wild-type sequence was used as input.

### Tokenization

In our tokenization scheme, we use the standard set of amino acids tokens as described in ESM-2 (ref. [7]). In addition to special tokens, the 20 wild-type amino acids tokens are as follows: D, N, E, K, V, Y, A, Q, M, I, T, L, R, F, G, C, S, P, H, W. We introduce new PTM tokens, corresponding to their unique specific UniProt annotations: `<N-linked (GlcNAc…) asparagine>`, `<Pyrrolidone carboxylic acid>`, `<Phosphoserine>`, `<Phosphothreonine>`, `<N-acetylalanine>`, `<N-acetylmethionine>`, `<N6-acetyllysine>`, `<Phosphotyrosine>`, `<S-diacylglycerol cysteine>`, `<N6-(pyridoxal phosphate)lysine>`, `<N-acetylserine>`, `<N6-carboxylysine>`, `<N6-succinyllysine>`, `<S-palmitoyl cysteine>`, `<O-(pantetheine 4-phosphoryl)serine>`, `<Sulfotyrosine>`, `<O-linked (GalNAc…) threonine>`, `<Omega-N-methylarginine>`, `<N-myristoyl glycine>`, `<4-hydroxyproline>`, `<Asymmetric dimethylarginine>`, `<N5-methylglutamine>`, `<4-aspartylphosphate>`, `<S-geranylgeranyl cysteine>`, `<4-carboxyglutamate>`. The top two most abundant PTM tokens are `<N-linked (GlcNAc…) asparagine>` and `<Phosphoserine>`. The full distribution of the PTM tokens is shown in Supplementary Fig. 1, and the full PTM tokens are presented in Supplementary Table 1. The wild-type amino acid tokens are then converted into embeddings by both ESM-2-650M and PTM-Mamba, while the PTM tokens are only processed by PTM-Mamba.

### PTM-Mamba training procedure

PTM-Mamba was trained on an Nvidia 8xA100 DGX system with 640 GB of shared VRAM on an adjusted masked language modeling task, in which, rather than random 15% token masking, we bias masked to PTM residue tokens (Fig. 1d). Briefly, given a sequence with 80% probability, we perform standard 15% token masking, and, with 20% probability, we mask all the PTM tokens and randomly mask 15% of wild-type tokens. For training, we then consider a protein sequence with masked residues, where the model aims to predict the original tokens at these residue positions. Let $x_i$ denote the original residue token at position $i$ that has been masked, and let $y_i$ denote the residue token predicted by the model for this position. The loss function $L$ for masked language modeling can be defined as the negative log likelihood of the correct tokens given their masked inputs, summed over all masked positions $N$:

$$L = -\sum_{i=1}^{N} \log P(x_i | x_{\text{masked}}).$$

$P(x_i | x_{\text{masked}})$ represents the probability of predicting the correct original token $x_i$ at the masked position, given the masked input sequence $x_{\text{masked}}$. PTM-Mamba was trained via the Adam optimizer with no weight decay. The final PTM-Mamba model has 24 layers with hidden dimensions of 768. It was trained for 16,765 steps (425 epochs) at a constant learning rate of 0.0002 with a batch size of 256 and dynamic batching. Training sequences were randomly cropped to a maximal length of 1,024 or padded at the end to reach a length of 1,024. During training, we clustered the sequences by length and constructed the batches. The training batches were fed into the model, going from the shortest to the longest sequences. PTM-Mamba was trained on an Nvidia 8xA100 GPU DGX system with 640 GB of shared VRAM.

### Benchmark model training

For all the benchmark tasks, we leverage the embeddings from pretrained PTM-Mamba and ESM-2 models and fine-tune a classification head on top of the embeddings. We extensively tuned the classification head architectures as well as the training hyperparameters for the benchmarks and have reported the optimal settings in Supplementary Codes 1–4 and Supplementary Table 2. For models trained on one-hot embeddings of wild-type input sequences, an nn.Embedding layer followed by a linear layer was used. All benchmark models were trained on an Nvidia 8xA100 GPU DGX system with 640 GB of shared VRAM. For robust performance comparison, we replicate each model ($n = 5$) and report the individual and average results. Models were evaluated using accuracy, precision, recall, $F_1$ score, MCC, AUROC and AUPRC metrics via scikit-learn[27].

### Reporting summary

Further information on research design is available in the Nature Portfolio Reporting Summary linked to this article.

## Data availability

All data needed to evaluate the conclusions are presented in the paper, tables and Supplementary Information and are further available at https://doi.org/10.5281/zenodo.14794992 (ref. [28]).

## Code availability

PTM-Mamba, PTM-Transformer, PTM-SaProt, baseline model weights, training code and Python scripts for data preprocessing can be found at https://huggingface.co/ChatterjeeLab/PTM-Mamba and https://github.com/programmablebio/ptm-mamba.

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

## Acknowledgements

We thank Mark III Systems for computing support. We further thank Y. Zhang and T. Chen for their insights related to the manuscript.

We thank L. Hong for rendering the PTM-Mamba logo. The work was supported by a grant from the National Institute of General Medical Sciences (award 1R35GM1555282-01) to the laboratory of P.C.

## Author contributions

F.Z.P. designed and implemented PTM-Mamba architecture and conducted benchmarking analysis. F.Z.P. and C.W. developed and trained PTM-SaProt. T.C., S.V. and B.S. conducted benchmarking analysis. F.Z.P., S.V., T.C. and P.C. wrote and reviewed the manuscript. P.C. conceived, designed, directed and supervised the study.

## Competing interests

The authors declare no competing interests.

## Additional information

**Correspondence and requests for materials** should be addressed to Pranam Chatterjee.

# Reporting Summary

## Statistics

For all statistical analyses, confirm that the following items are present in the figure legend, table legend, main text, or Methods section.

| n/a | Confirmed | |
|---|---|---|
| ☐ | ☒ | The exact sample size (*n*) for each experimental group/condition, given as a discrete number and unit of measurement |
| ☐ | ☒ | A statement on whether measurements were taken from distinct samples or whether the same sample was measured repeatedly |
| ☒ | ☐ | The statistical test(s) used AND whether they are one- or two-sided<br>*Only common tests should be described solely by name; describe more complex techniques in the Methods section.* |
| ☒ | ☐ | A description of all covariates tested |
| ☒ | ☐ | A description of any assumptions or corrections, such as tests of normality and adjustment for multiple comparisons |
| ☐ | ☒ | A full description of the statistical parameters including central tendency (e.g. means) or other basic estimates (e.g. regression coefficient) AND variation (e.g. standard deviation) or associated estimates of uncertainty (e.g. confidence intervals) |
| ☒ | ☐ | For null hypothesis testing, the test statistic (e.g. *F*, *t*, *r*) with confidence intervals, effect sizes, degrees of freedom and *P* value noted<br>*Give P values as exact values whenever suitable.* |
| ☒ | ☐ | For Bayesian analysis, information on the choice of priors and Markov chain Monte Carlo settings |
| ☒ | ☐ | For hierarchical and complex designs, identification of the appropriate level for tests and full reporting of outcomes |
| ☒ | ☐ | Estimates of effect sizes (e.g. Cohen's *d*, Pearson's *r*), indicating how they were calculated |

*Our web collection on statistics for biologists contains articles on many of the points above.*

## Software and code

Policy information about availability of computer code

| Data collection | Model training data was curated from UniProt15. Specifically, 311,350 experimentally-validated PTM records were collected from Swiss-Prot and mapped the PTM annotations of their protein to their respective sequences to construct the new PTM sequences. The final dataset includes a total of 79,707 PTM sequences. Data curation code can be found at: https://github.com/programmablebio/ptm-mamba/tree/main/ptm_data_preprocessing.<br><br>Datasets for the four benchmarking tasks were collected from the following sources. Phosphorylation site data was obtained from the corresponding ProteinBERT benchmark,25 originally derived from PhospoSitePlus18 and filtered for sequences between 256 and 512 amino acids in length, yielding a training set of 15,588 sequences, a validation set of 1707 sequences, and a testing set of 3106 sequences. Non-histone acetylation site prediction was performed equivalently as described in prior literature, using the NHAC (Non-Histone Acetylation Collection) dataset.26 The druggability and disease association datasets were curated from the dbPTM database.17 PPI data describing the effect of PTMs were curated from PTMint, which encompasses 2,477 non-redundant PTM sites in 1169 proteins affecting 2371 protein-protein pairs. Briefly, wild-type sequences were mapped to corresponding entries in the PTM-Mamba dataset, and wild-type residues were replaced by the corresponding, position-specific PTMs for tokenization by specified models. For all other baseline models trained with standard one-hot embeddings or ESM-2 embeddings, the corresponding wild-type sequence was used as input. |
|---|---|
| Data analysis | For all the benchmark tasks, we leverage the embeddings from pretrained PTM-Mamba and ESM-2 models and finetune a classification head on top of the embeddings. We extensively tuned the classification head architectures as well as the training hyperparameters for the benchmarks and have reported the optimal settings in Supplementary Listing 1-4 and Supplementary Table 2. For models trained on one-hot embeddings of wild-type input sequences, an nn.Embedding layer followed by a linear layer was utilized. All benchmark models were trained on a Nvidia 8xA100 GPU DGX system with 640 GB of shared VRAM. For robust performance comparison, we replicate each model (n=5) and |

report the individual and average results. Models were evaluated using the Accuracy, Precision, Recall, F1 Score, MCC, AUROC, and AUPRC metrics via scikit-learn.27

For manuscripts utilizing custom algorithms or software that are central to the research but not yet described in published literature, software must be made available to editors and reviewers. We strongly encourage code deposition in a community repository (e.g. GitHub). See the Nature Portfolio guidelines for submitting code & software for further information.

## Data

Policy information about availability of data

All manuscripts must include a data availability statement. This statement should provide the following information, where applicable:
- Accession codes, unique identifiers, or web links for publicly available datasets
- A description of any restrictions on data availability
- For clinical datasets or third party data, please ensure that the statement adheres to our policy

All data needed to evaluate the conclusions are presented in the paper, tables, and supplementary information, and are further available at https://doi.org/10.5281/zenodo.14794992.

## Human research participants

Policy information about studies involving human research participants and Sex and Gender in Research.

| Reporting on sex and gender | N/A |
| Population characteristics | N/A |
| Recruitment | N/A |
| Ethics oversight | N/A |

Note that full information on the approval of the study protocol must also be provided in the manuscript.

# Field-specific reporting

Please select the one below that is the best fit for your research. If you are not sure, read the appropriate sections before making your selection.

☒ Life sciences      ☐ Behavioural & social sciences      ☐ Ecological, evolutionary & environmental sciences

For a reference copy of the document with all sections, see nature.com/documents/nr-reporting-summary-flat.pdf

# Life sciences study design

All studies must disclose on these points even when the disclosure is negative.

| Sample size | Sample size determination was not established. We chose sample sizes based on data availability and number of models required for benchmarking. |
| Data exclusions | No data was excluded. |
| Replication | All benchmarks in panels A-C were performed in replicates (n = 5). All attempts at replication were successful. |
| Randomization | Random seeds were used to prevent overfitting. |
| Blinding | No blinding was necessary for this study. As a computational study, the training models must be known to the researcher to accurately evaluate results. |

# Reporting for specific materials, systems and methods

We require information from authors about some types of materials, experimental systems and methods used in many studies. Here, indicate whether each material, system or method listed is relevant to your study. If you are not sure if a list item applies to your research, read the appropriate section before selecting a response.

## Materials & experimental systems

| n/a | Involved in the study |
|-----|----------------------|
| ☒ | ☐ Antibodies |
| ☒ | ☐ Eukaryotic cell lines |
| ☒ | ☐ Palaeontology and archaeology |
| ☒ | ☐ Animals and other organisms |
| ☒ | ☐ Clinical data |
| ☒ | ☐ Dual use research of concern |

## Methods

| n/a | Involved in the study |
|-----|----------------------|
| ☒ | ☐ ChIP-seq |
| ☒ | ☐ Flow cytometry |
| ☒ | ☐ MRI-based neuroimaging |

