## [Peer Review File · Nature Methods]

PTM-Mamba: A PTM-Aware Protein Language Model with Bidirectional Gated Mamba Blocks

Corresponding Author: Professor Pranam Chatterjee

Version 0:

Decision Letter:

22nd May 2024

Dear Pranam,

Your Article, "PTM-Mamba: A PTM-Aware Protein Language Model with Bidirectional Gated Mamba Blocks", has now been seen by 2 reviewers. As you will see from their comments below, although the reviewers find your work of considerable potential interest, they have raised a number of concerns. We are interested in the possibility of publishing your paper in Nature Methods, but would like to consider your response to these concerns before we reach a final decision on publication.

We therefore invite you to revise your manuscript to address these concerns. In particular, we would like you to address all the technical points and suggestion made by the reviewers. We would also like you to be as detailed with the method description as possible. Finally, we would like you to refer to this editorial where we talk about how the base methods should be published in a peer reviewed journal before other methods build upon them, and the generated results are published (<https://www.nature.com/articles/s41592-020-01017-y>). We are concerned that the Mamba paper (<https://arxiv.org/abs/2312.00752>) has not been published in a peer review journal yet, although we understand that this isn't under your control.

Link Redacted

We hope to receive your revised paper within 8 weeks. If you cannot send it within this time, please let us know. In this event, we will still be happy to reconsider your paper at a later date so long as nothing similar has been accepted for publication at

Nature Methods or published elsewhere.

OPEN SCIENCE REQUIREMENTS

REPORTING SUMMARY AND EDITORIAL POLICY CHECKLISTS

DATA AVAILABILITY

All novel DNA and RNA sequencing data, protein sequences, genetic polymorphisms, linked genotype and phenotype data, gene expression data, macromolecular structures, and proteomics data must be deposited in a publicly accessible database, and accession codes and associated hyperlinks must be provided in the "Data Availability" section.

CODE AVAILABILITY

Please include a "Code Availability" subsection in the Online Methods which details how your custom code is made available. Only in rare cases (where code is not central to the main conclusions of the paper) is the statement "available upon request" allowed (and reasons should be specified).

MATERIALS AVAILABILITY

As a condition of publication in Nature Methods, authors are required to make unique materials promptly available to others

without undue qualifications.

SUPPLEMENTARY PROTOCOL

To help facilitate reproducibility and uptake of your method, we ask you to prepare a step-by-step Supplementary Protocol for the method described in this paper. We [encourage authors to share their step-by-step experimental protocols](https://www.nature.com/nature-research/editorial-policies/reporting-standards#protocols) on a protocol sharing platform of their choice and report the protocol DOI in the reference list. Nature Portfolio's protocols.io is a free-to-use and open resource for protocols; protocols deposited onto protocols.io are citable and can be linked from the published article. More details can found at [protocols.io](https://www.protocols.io/help/publish-articles).

ORCID

Sincerely,
Arunima

Arunima Singh, Ph.D.
Senior Editor
Nature Methods

Reviewers' Comments:

Reviewer #2:

Remarks to the Author:

The authors compile a training set of protein sequences with post-translational modifications (PTMs). They then extend the amino acid alphabet to include PTMs and train a mamba-based addition to ESM-2 that learning to reconstruct partially masked sequences of post-translationally-modified proteins, which they call PTM-Mamba. They show that the representations satisfy biochemical intuitions and that models trained on top of PTM-Mamba are better at downstream tasks involving PTMs than ESM-2 and other baselines.

Strengths

The authors identify an important domain that has not yet received much attention from the machine-learning community and propose an intuitive and apparently effective pretraining method.

The bidirectional mamba architecture is simple, well-described, and could be useful for many other applications.

Performance on downstream tasks is generally strong.

Weaknesses and questions

The Bidirectional Mamba architecture described in Listing 1 and in their Github only works if the inputs are always of the same length so that a sequence always has the same amount of padding. This means that clustering the sequences by length to construct batches doesn't provide any benefit, as the shorter sequences still must be padded to 1024, and thus there is no improvement in training throughput, while the non-random sampling introduces some bias. At inference time, this means the model cannot be easily extended to longer sequences, and that shorter sequences must be padded to 1024, increasing computational cost. This could be fixed by writing a custom flip function that only flips the unpadded portion of the reverse sequence such that the output is not dependent on the amount of padding. This has the added benefit of making it so that representations for the same location are always concatenated together.

For example, here is some code to generate the indices for flipping the reverse sequences:

```
def get_flip_inds(hidden_states, lengths):
# hidden_states: b x ell x d
# lengths: b x 1
b, ell, d = hidden_states.shape
device = hidden_states.device
xi = torch.arange(b, device=device).repeat_interleave(ell, dim=0).view(b, ell)
yi = (torch.arange(ell, device=device) + (ell - lengths)) % ell
return xi, yi
```

then you could do `flipped = bwd[xi, yi]` to flip just the non-padding portions of the backwards hidden states.

It's not described how or why masked language model pretraining should benefit these particular downstream tasks. For many protein tasks, previous work shows that the language model pretraining primarily serves to separate functional and non-functional sequence variants. Does this apply also to PTMs? If not, is there another mechanism by which pretraining should help? This is of special concern for the druggability and disease association datasets.

While most of the datasets and downstream tasks are described in sufficient detail, the paper needs to specifically define what disease association is. It would also be helpful to have an idea of how large and diverse this dataset is and how it was collected. This may also make it clearer why PTM pretraining improves performance.

It's not clear whether the tests sets for the phosphorylation and acetylation downstream tasks are deduplicated against the pretraining dataset. If not, the improvement can be trivially attributed to leakage from the pretraining.

How were the 311,350 SwissProt sequences down-selected to 79,707?

For the downstream tasks, the conclusions would be more robust with replicates and uncertainty estimates on the reported metrics.

The manuscript also implies that there was only one pretraining epoch: is this correct?

In the introduction, the authors state that Mamba uses "hardware-aware primitives to overcome the quadratic time complexity of the standard attention mechanism." This isn't quite right: the selective state-space architecture itself gives sub-quadratic time and memory complexity with sequence length. The hardware-aware primitives accelerate the computations but do not affect scaling with sequence length.

In the section on "zero-shot PTM discovery," what does are "randomly downselected wild-type sequences?"

The paper should cite scikit-learn, given that it is explicitly references.

I think the list of tokens would be clearer as a table. I do commend the authors for putting the entire list in the manuscript though, as it makes everything much clearer and more reproducible.

Why does ESM-2-3B performance vary much more with length than the other models?

Reviewer #3:

Remarks to the Author:

The core contribution of this paper is the introduction of a novel protein language model, PTM-Mamba, which distinguishes itself from previous language models such as ESM and ProtTrans by explicitly considering post-translational modifications (PTMs). The authors integrate Mamba, a new model architecture, into the existing language models ESM by replacing wild-type amino acids with new PTM tokens. From the perspective of novelty in the paper, I think that it is the first study to address the issue of PTMs and effectively model them, making it a valuable academic contribution. Additionally, the paper showcases several advantages of PTM-Mamba in the multiple tasks.

There are also a few questions that need to be addressed.

First, have the authors compared the performance of PTM-Mamba with a Transformer architecture to evaluate if the switch to Mamba is justified, or is it solely because Mamba has gained popularity in the NLP and CV communities? Second, although PTM-Mamba demonstrates significant advantages in downstream tasks related to PTMs, how does it perform on common protein benchmarks such as zero-shot mutation effect prediction (e.g., the ClinVar dataset in EVE [1]) and ProteinGym leaderboard? Third, the paper introduces a new vocabulary with PTM tokens, I found some recent literature has also proposed enhanced or new vocabularies, such as Saprot[2], which combines amino acids with 3Di tokens and has shown superior performance compared to ESM-2 in certain tasks. It would be worthwhile for the authors to consider comparing their approach with Saprot, especially since they combine ESM with PTM-Mamba. If ESM is replaced with Saprot, would the performance improvements observed in the paper still hold? This would help evaluate the generalizability of the proposed method, or does Saprot incorporating structural information might implicitly contain PTM information. It would be interesting to see such results.

Lastly, I cannot find the details of the baseline and hyperparameter settings used for comparisons in the paper. Hyperparameter settings are crucial in machine learning algorithms, and it would be beneficial for the authors to provide an explanation of their chosen settings and present the results accordingly. We need fair comparison when claiming a methodology works.

While the experiments in this paper validate PTM-Mamba from several perspectives, the depth and biological insights seem insufficient for an Article paper in Nature Methods. I would suggest that the paper be published as a brief communication, which often emphasizes its broad interest and impact. Future research can continue to explore and validate PTM-Mamba based on this foundation, and it is highly anticipated that PTM-Mamba will assist biologists in their work.

Version 1:

Decision Letter:

Our ref: NMETH-BC55553A

24th Dec 2024

Dear Pranam,

Thank you for submitting your revised manuscript "PTM-Mamba: A PTM-Aware Protein Language Model with Bidirectional Gated Mamba Blocks" (NMETH-BC55553A). It has now been seen by the original referees and their comments are below. The reviewers find that the paper has improved in revision, and therefore we'll be happy in principle to publish it in Nature Methods, pending minor revisions to satisfy the referees' final requests and to comply with our editorial and formatting guidelines.

TRANSPARENT PEER REVIEW

ORCID

Sincerely,
Arunima

Arunima Singh, Ph.D.
Senior Editor
Nature Methods

Reviewer #2 (Remarks to the Author):

I've looked at the revisions and rebuttals. For the most part, the authors have done a good job of addressing the reviewer concerns. I disagree with their assessment that doing the flipping properly would be too slow (yes, it's slower than the naive flip, but it's a tiny fraction of the training time), but the overall results of the paper are strong.

Reviewer #3 (Remarks to the Author):

Thank you for your detailed explanation and revision point by point. I have read the author's comments carefully and my key technical concerns have been addressed. Please put your new results into the manuscript. In addition, the authors should pay more attention to their presentation. The current format is not suitable for brief communication. For such a paper, it is crucial to succinctly summarize the key novelties, technical contributions, and main results. The current version is not quite consistent with the expected format and needs further improvement.

Reviewer #3 (Remarks on code availability):

No additional comments

Version 2:

Decision Letter:

5th Mar 2025

Dear Pranam,

I am pleased to inform you that your Brief Communication, "PTM-Mamba: A PTM-Aware Protein Language Model with Bidirectional Gated Mamba Blocks", has now been accepted for publication in Nature Methods. The received and accepted dates will be February 27, 2024 and March 5, 2025. This note is intended to let you know what to expect from us over the next month or so, and to let you know where to address any further questions.

Over the next few weeks, your paper will be copyedited to ensure that it conforms to Nature Methods style. Once your paper is typeset, you will receive an email with a link to choose the appropriate publishing options for your paper and our Author Services team will be in touch regarding any additional information that may be required.

Once proofs are generated, they will be sent to you electronically and you will be asked to send a corrected version within 48 hours. It is extremely important that you let us know now whether you will be difficult to contact over the next month. If this is the case, we ask that you send us the contact information (email, phone and fax) of someone who will be able to check the proofs and deal with any last-minute problems.

If, when you receive your proof, you cannot meet the deadline, please inform us at rjsproduction@springernature.com immediately.

If you are active on Twitter/X, please e-mail me your and your coauthors' handles so that we may tag you when the paper is published.

To assist our authors in disseminating their research to the broader community, our SharedIt initiative provides you with a unique shareable link that will allow anyone (with or without a subscription) to read the published article. Recipients of the link with a subscription will also be able to download and print the PDF. As soon as your article is published, you will receive an automated email with your shareable link.

Please note that you and your coauthors may order reprints and single copies of the issue containing your article through

Springer Nature Limited's reprint website, which is located at <http://www.nature.com/reprints/author-reprints.html>. If there are any questions about reprints please send an email to author-reprints@nature.com and someone will assist you.

Best regards,
Arunima

Arunima Singh, Ph.D.
Senior Editor
Nature Methods

** Visit the Springer Nature Editorial and Publishing website at http://www.springernature.com/editorial-and-publishing-jobs?utm_source=ejP_NMeth_email&utm_medium=ejP_NMeth_email&utm_campaign=ejp_Nmeth for more information about our career opportunities. If you have any questions please click [here](mailto:editorial.publishing.jobs@springernature.com). **

Open Access This Peer Review File is licensed under a Creative Commons Attribution 4.0 International License, which permits use, sharing, adaptation, distribution and reproduction in any medium or format, as long as you give appropriate credit to the original author(s) and the source, provide a link to the Creative Commons license, and indicate if changes were made. In cases where reviewers are anonymous, credit should be given to 'Anonymous Referee' and the source.

NMETH-A55553 - Response To Reviewers

Reviewer 2:

“The Bidirectional Mamba architecture described in Listing 1 and in their Github only works if the inputs are always of the same length so that a sequence always has the same amount of padding. This means that clustering the sequences by length to construct batches doesn't provide any benefit, as the shorter sequences still must be padded to 1024, and thus there is no improvement in training throughput, while the non-random sampling introduces some bias. At inference time, this means the model cannot be easily extended to longer sequences, and that shorter sequences must be padded to 1024, increasing computational cost. This could be fixed by writing a custom flip function that only flips the unpadded portion of the reverse sequence such that the output is not dependent on the amount of padding. This has the added benefit of making it so that representations for the same location are always concatenated together.

For example, here is some code to generate the indices for flipping the reverse sequences:

```
def get_flip_inds(hidden_states, lengths):  
# hidden_states: b x ell x d  
# lengths: b x 1  
b, ell, d = hidden_states.shape  
device = hidden_states.device  
xi = torch.arange(b, device=device).repeat_interleave(ell, dim=0).view(b, ell)  
yi = (torch.arange(ell, device=device) + (ell - lengths)) % ell  
return xi, yi
```

then you could do `flipped = bwd[xi, yi]` to flip just the non-padding portions of the backwards hidden states.”

We thank the reviewer for their valuable suggestion regarding the custom flip function to flip only the non-padding tokens, which indeed provides a more precise bidirectional information fusion. However, we found that this approach is significantly slower than the regular flip operation. In our efficiency benchmark (see table below), we compared the custom flip function with our default flip function (`torch.flip` from PyTorch). The results show that, on average, the custom flip function is three times slower than the default one, which would prolong the training time threefold. We appreciate the reviewer's insight and will consider this trade-off in future developments to balance precision and computational efficiency. Please find the results and code below for the reviewer's purposes.

Batch Size	Sequence Length	Custom Flip Time	Regular Flip Time	Custom Time / Regular Time
16	64	0.00742	0.00247	3.0027
16	128	0.00287	0.00081	3.54149
16	512	0.0075	0.0025	2.99803
16	1024	0.00752	0.00251	2.99871
32	64	0.00596	0.00198	3.00867
32	128	0.00747	0.00248	3.01687
32	512	0.00751	0.0025	3.00891
32	1024	0.00756	0.00253	2.99431
360	64	0.00618	0.00205	3.01295
360	128	0.00764	0.00256	2.97858
360	512	0.00809	0.00282	2.86995
360	1024	0.00881	0.00319	2.76468

Average		0.00705	0.00237	3.01063
---------	--	---------	---------	---------

Supplement. Python code that benchmarks the custom flip function with the default flip function in the pytorch cuda setting.

```

import torch
import time
import pandas as pd

def get_flip_inds(hidden_states, lengths):
    # hidden_states: b x ell x d
    # lengths: b x 1
    b, ell, d = hidden_states.shape
    device = hidden_states.device
    xi = torch.arange(b, device=device).repeat_interleave(ell, dim=0).view(b, ell)
    yi = (torch.arange(ell, device=device) + (ell - lengths)) % ell
    return xi, yi

def benchmark_flip_functions(batch_size, sequence_length, hidden_dim, num_runs=10):
    device = torch.device("cuda" if torch.cuda.is_available() else "cpu")

    # Random input data
    hidden_states = torch.randn(batch_size, sequence_length, hidden_dim).to(device)
    lengths = torch.randint(1, sequence_length + 1, (batch_size, 1)).to(device)

    # Warm-up to ensure GPU is ready
    xi, yi = get_flip_inds(hidden_states, lengths)
    flipped_custom = hidden_states[xi, yi]
    flipped_regular = torch.flip(hidden_states, [1])
    torch.cuda.synchronize(device)

    custom_times = []
    regular_times = []

    for _ in range(num_runs):
        # Measure time for get_flip_inds
        start_time = time.time()
        xi, yi = get_flip_inds(hidden_states, lengths)
        flipped_custom = hidden_states[xi, yi]
        torch.cuda.synchronize(device)
        custom_times.append(time.time() - start_time)

        # Measure time for regular flip
        start_time = time.time()
        flipped_regular = torch.flip(hidden_states, [1])
        torch.cuda.synchronize(device)
        regular_times.append(time.time() - start_time)

    custom_time = sum(custom_times) / num_runs
    regular_time = sum(regular_times) / num_runs

    return custom_time, regular_time

# Test parameters
batch_sizes = [16, 32, 360]
sequence_lengths = [64, 128, 512, 1024]
hidden_dim = 380
num_runs = 10

results = []

for batch_size in batch_sizes:

```

```
for sequence_length in sequence_lengths:
    custom_time, regular_time = benchmark_flip_functions(batch_size, sequence_length, hidden_dim, num_runs)
    ratio = custom_time / regular_time
    results.append((batch_size, sequence_length, custom_time, regular_time, ratio))

df_results = pd.DataFrame(results, columns=["Batch Size", "Sequence Length", "Custom Flip Time", "Regular Flip Time", "Custom Time / Regular Time"])

df_results = df_results.round(5)
df_results
```

"It's not described how or why masked language model pretraining should benefit these particular downstream tasks. For many protein tasks, previous work shows that the language model pretraining primarily serves to separate functional and non-functional sequence variants. Does this apply also to PTMs? If not, is there another mechanism by which pretraining should help? This is of special concern for the druggability and disease association datasets."

We thank the reviewer for their insightful question regarding the benefits of masked language model pretraining for downstream tasks, particularly in the context of post-translational modifications (PTMs), druggability, and disease association datasets. Recent studies have demonstrated that embeddings derived from protein language models (pLMs) extend beyond distinguishing functional from non-functional sequence variants; they also capture intricate functional features of proteins. For instance, most benchmarking have shown that pLM embeddings can effectively predict protein secondary structures, subcellular localization, and even functional annotations without relying on evolutionary information (10.48550/arXiv.2007.06225, 10.1101/2024.06.03.597245). Additionally, fine-tuning these embeddings has been shown to enhance their utility in functional similarity evaluations, indicating their adaptability to various protein-related tasks (10.1093/bib/bbad117). In the context of PTMs, druggability, and disease associations, the rich representations learned during pretraining via MLM enable models to identify subtle sequence patterns and structural motifs associated with these specific functions. This capability is particularly beneficial for tasks where traditional sequence alignment methods may fall short, as the embeddings can capture context-dependent information that is crucial for understanding protein function and interaction. Therefore, we are confident that the pretraining phase equips the PTM-Mamba with a comprehensive understanding of protein language, which can be fine-tuned to address the complexities of PTM effects, including druggability and disease associations, ultimately enhancing predictive performance in these areas.

To directly address this question, our results indeed support that pretraining significantly improves performance on these tasks. By comparing models with and without PTM embeddings, we observe that adding ESM embeddings, followed by PTM-Mamba, provides substantial gains for both druggability and disease association predictions. Specifically, our results show that the pretrained ESM-2 model, further fine-tuned with PTM-Mamba, outperforms both one-hot encoding and one-hot with PTM information. This indicates that pretraining contributes meaningfully to performance improvements, especially in complex predictive tasks related to druggability and disease associations.

"While most of the datasets and downstream tasks are described in sufficient detail, the paper needs to specifically define what disease association is. It would also be helpful to have an idea of how large and diverse this dataset is and how it was collected. This may also make it clearer why PTM pretraining improves performance"

We thank the reviewer for pointing out our lack of detail here, and giving us an opportunity to explain. Disease association (derived from dbPTM) denotes the connection of PTMs with various diseases, including heart disease, cancer, neurodegenerative diseases, and diabetes, with 2,846 disease-associated PTM sites identified. For disease association labels, data was sourced from databases like PhosphoSitePlus, ActiveDriverDB, and Genome-Wide Association Studies (GWAS), along with manual curation of scientific literature. The process included systematically querying PTM-related keywords in research articles, followed by manual extraction of PTM sites linked to diseases. To enhance transparency, we have made all the benchmark datasets available on Google Drive.

"It's not clear whether the tests sets for the phosphorylation and acetylation downstream tasks are deduplicated against the pretraining dataset. If not, the improvement can be trivially attributed to leakage from the pretraining."

We acknowledge the reviewer’s potential concern for data leakage between the pretraining and downstream tasks. To quantify the data leakage between the pretraining PTM data and the downstream test set, we use the mmseq2 to calculate the pairwise sequence identity between the two sets and plot the distribution as shown below (the Python implementation is available at Google Drive). While we observe a few highly similar (sequence identity > 90%), 72% of the pairs are nonhomologous (sequence identity <50%). Since ESM was trained on UniProt sequences, it also suffers from this data leakage issue.

Additionally, we would like to clarify the following points to demonstrate that the sequences, while similar, are not identical. The downstream tasks use wild type sequences, which do not align directly with the masked PTM sequences used during pre-training as they are two different token representations in essence. The divergence in input types ensures that the model does not leverage direct sequence matches but rather the generalized knowledge acquired during pretraining. Second, the pretraining and downstream tasks possess **two different learning objectives**. The pretraining task involves a classification targeting PTMs and the 20 types of amino acids. In contrast, the downstream tasks focus on binary classification specific to a type of PTM (phosphorylation or acetylation). The fundamental objectives between these tasks differ, leading to distinct learning outcomes. We admit that the model indeed learns the semantics of PTMs during pretraining, which aids in the fine-tuning process for specific PTM tasks. This transfer of knowledge does not indicate leakage but rather effective knowledge transfer.

“How were the 311,350 SwissProt sequences down-selected to 79,707?”

We thank the reviewer for their question regarding the down-selection process of the 311,350 SwissProt sequences to 79,707. In UniProt, a single protein record can have multiple PTM annotations. During our curation process, we iterated through all the protein records in SwissProt along with their PTM annotations, which resulted in 311,350 PTM labels for all the protein sequences. We then mapped these annotations back to the sequences, yielding a total of 79,707 PTM sequences. To promote reproducibility, we have open-sourced the data curation code at: https://github.com/programmablebio/ptm-mamba/tree/main/ptm_data_preprocessing.

“For the downstream tasks, the conclusions would be more robust with replicates and uncertainty estimates on the reported metrics.”

We thank the reviewer for highlighting the importance of replicates and uncertainty estimates for robust performance comparisons in our downstream tasks. To address this, we have conducted 5-fold cross-validation experiments across our existing downstream tasks, as well as an additional benchmark focused on PTM-related functionality. These 5-fold experiments provide more robust performance comparisons, demonstrating that PTM-Mamba consistently outperforms both the ESM series of models and one-hot baselines across most metrics. Notably, our added benchmark—predicting

the effect of PTMs on protein-protein interaction (PPI)—specifically leverages PTM tokens, providing a meaningful test of PTM-Mamba’s capabilities. Our model performs strongly on this PTM-relevant benchmark, underscoring the advantage of incorporating PTM information in predictive tasks. While the 5-fold experiments show that PTM-Mamba maintains strong overall performance, they also reveal slight variations in certain metrics compared to initial results, as expected. These refinements offer a more comprehensive view of the model’s strengths and reinforce the robustness of PTM-Mamba across diverse PTM-focused and traditional benchmarks. Updated results can be found in Figure 2 of the manuscript.

“The manuscript also implies that there was only one pretraining epoch: is this correct?”

We trained PTM-Mamba for 425 epochs (16,765 iterations) until the validation loss converges as shown in the val_loss curve below. These details are now included in the methods section.

“In the introduction, the authors state that Mamba uses “hardware-aware primitives to overcome the quadratic time complexity of the standard attention mechanism.” This isn’t quite right: the selective state-space architecture itself gives sub-quadratic time and memory complexity with sequence length. The hardware-aware primitives accelerate the computations but do not affect scaling with sequence length. “

Thank you for pointing out the inaccuracy in our description. We have revised the introduction to clarify that the selective state-space architecture provides sub-quadratic time and memory complexity with sequence length, while the hardware-aware primitives accelerate computations without affecting scaling. We appreciate your insightful feedback.

“In the section on “zero-shot PTM discovery,” what does are “randomly downselected wild-type sequences?””

We appreciate the reviewer’s feedback on the ambiguity of our statement. For the zero-shot PTM discovery, we randomly chose a set of regular protein sequences, and ran the sequences through PTM-Mamba. These were sequences not found in the training set of PTM-Mamba. We then observed the output logits over the entire PTM vocabulary and observed that PTMs that can be installed on specific amino acids were actually enriched at those positions in a zero-shot manner. To make the logic more clear, we open the code at the official GitHub repository.

“The paper should cite scikit-learn, given that it is explicitly references.”

We have added the scikit-learn reference as follows. “Models were all trained for 40 epochs using one Nvidia A100 GPU with 80 GB of VRAM. Models were evaluated using the Accuracy, Precision, Recall, F1 Score, MCC, AUROC, and AUPRC metrics via scikit-learn [Pedregosa et al., 2011].”

“I think the list of tokens would be clearer as a table. I do commend the authors for putting the entire list in the manuscript though, as it makes everything much clearer and more reproducible.”

To make the PTM encoding clear, we describe the input PTM tokens in Supplementary Table 1 as shown below.

PTM	Annotation
N-linked (GlcNAc...) asparagine	Addition of N-acetylglucosamine to asparagine residues
Pyrrolidone carboxylic acid	Formation of a lactam ring from the N-terminal glutamine
Phosphoserine	Addition of a phosphate group to serine residues
Phosphothreonine	Addition of a phosphate group to threonine residues
N-acetylalanine	Acetylation of the N-terminal alanine
N-acetylmethionine	Acetylation of the N-terminal methionine
N6-acetyllysine	Acetylation of the lysine residue at the -amino group
Phosphotyrosine	Addition of a phosphate group to tyrosine residues
S-diacylglycerol cysteine	Attachment of diacylglycerol to cysteine residues
N6-(pyridoxal phosphate)lysine	Addition of pyridoxal phosphate to lysine residues
N-acetyserine	Acetylation of the serine residue
N6-carboxyllysine	Carboxylation of the lysine residue
N6-succinyllysine	Succinylation of the lysine residue
S-palmitoyl cysteine	Palmitoylation of cysteine residues
O-(pantetheine 4-phosphoryl)serine	Addition of pantetheine phosphate to serine residues
Sulfotyrosine	Sulfation of tyrosine residues
O-linked (GalNAc...) threonine	Addition of N-acetylgalactosamine to threonine residues
Omega-N-methylarginine	Methylation of the arginine residue
N-myristoyl glycine	Myristoylation of glycine residues
4-hydroxyproline	Hydroxylation of proline residues
Asymmetric dimethylarginine	Dimethylation of the arginine residue in an asymmetric manner
N5-methylglutamine	Methylation of the glutamine residue
4-aspartylphosphate	Addition of a phosphate group to aspartate residues
S-geranylgeranyl cysteine	Attachment of geranylgeranyl to cysteine residues
4-carboxyglutamate	Carboxylation of glutamate residues

Table 3: PTM tokens that PTM-Mamba encodes and their Annotations.

“Why does ESM-2-3B performance vary much more with length than the other models?”

We acknowledge the author's observation regarding ESM-2-3B's performance variability with length, and agree that, given its size, ESM-2-3B is expected to perform strongly. Our 5-fold experiments indicate that ESM2-3B performs comparably to ESM2-650M. In some benchmarks and across certain metrics, ESM2-3B outperforms, while in others, ESM2-650M performs better. For instance, in Disease-association prediction, ESM2-3B shows stronger performance in terms of AUROC and AUPRC and slightly worse in Accuracy, F1, MCC etc. However, we do not believe these differences are statistically significant.

In addition, the variation in performance of the ESM-2-3B model has been observed in multiple studies, including our previous ones (Bhat, et al. 2023 *bioRxiv*, and Brix, et al. 2023 *Communications Biology*). In many instances, ESM-2-3B shows degraded performance compared to the ESM-2-650M model. Unlike autoregressive models, where scaling laws typically result in performance improvements with an increase in parameters, bidirectional models such as BERT tend to saturate at a certain scale, resulting in negligible improvements beyond that point. Additionally, the hyperparameter search for the ESM-2-3B model has not been as comprehensive as for the ESM-2-650M model due to the expensive training costs involved, leading to suboptimal optimization. For example, SaProt demonstrates that the ESM-2-15B model does not outperform its 650M version in zero-shot mutation prediction, indicating that simply increasing model size does not guarantee better performance (Table 6 Su, et al. 2023 *bioRxiv*).

Reviewer 3:

“The core contribution of this paper is the introduction of a novel protein language model, PTM-Mamba, which distinguishes itself from previous language models such as ESM and ProtTrans by explicitly considering post-translational modifications (PTMs). The authors

integrate Mamba, a new model architecture, into the existing language models ESM by replacing wild-type amino acids with new PTM tokens. From the perspective of novelty in the paper, I think that it is the first study to address the issue of PTMs and effectively model them, making it a valuable academic contribution. Additionally, the paper showcases several advantages of PTM-Mamba in the multiple tasks.”

We thank the reviewer for the kind comments!

“First, have the authors compared the performance of PTM-Mamba with a Transformer architecture to evaluate if the switch to Mamba is justified, or is it solely because Mamba has gained popularity in the NLP and CV communities?”

The primary motivation for using Mamba over a traditional Transformer architecture lies in its computational efficiency and reduced memory usage, making training and inference more accessible to a broader range of users. Below, we present the training curves of PTM-Mamba and a PTM-Transformer model of similar size (220M parameters). Our results indicate that PTM-Mamba converges faster during training. Moreover, due to its subquadratic computational complexity, PTM-Mamba uses less memory and exhibits faster inference times compared to the Transformer architecture.

“Second, although PTM-Mamba demonstrates significant advantages in downstream tasks related to PTMs, how does it perform on common protein benchmarks such as zero-shot mutation effect prediction (e.g., the ClinVar dataset in EVE [1]) and ProteinGym leaderboard?”

We sincerely thank the reviewer for these valuable suggestions regarding additional benchmark evaluations, such as zero-shot mutation effect prediction with the ClinVar dataset in EVE and the ProteinGym leaderboard. Our focus with PTM-Mamba has been on PTM-related tasks, given the model's specialized design to tokenize and capture post-translational modification features. Benchmarks like zero-shot mutation effect prediction and ProteinGym are highly informative for many applications, yet they are not directly aligned with PTM-specific modeling. However, we appreciate the importance of broader benchmark comparisons, and we look forward to exploring these types of evaluations in future work. This will allow us to further characterize PTM-Mamba's strengths across different tasks, providing additional insights into its versatility and potential applications. We have, however, added a new benchmark: PTM effect on PPIs, which can benefit from the introduction of PTM tokens, unlike the common benchmarks. Our results demonstrate strong performance on this task compared to all other models.

“Third, the paper introduces a new vocabulary with PTM tokens, I found some recent literature has also proposed enhanced or new vocabularies, such as SaProt[2], which combines amino acids with 3Di tokens and has shown superior performance compared to ESM-2 in certain tasks. It would be worthwhile for the authors to consider comparing their approach with SaProt, especially since they combine ESM with PTM-Mamba. If ESM is replaced with SaProt, would the performance improvements observed in the paper still hold? This would help evaluate the generalizability of the proposed method, or does SaProt incorporating structural information might implicitly contain PTM information. It would be interesting to see such results.”

Thank you for the insightful comment regarding the use of SaProt and its potential for comparison with our approach. In response, we introduced a new model variant, **PTM-SaProt**, which leverages the Mamba architecture while incorporating structural-aware embeddings from SaProt instead of ESM2 embeddings. For PTM-SaProt pretraining, we pre-folded the

structures of all PTM sequences (ranging from lengths of 20 to 1024) using ESMFold, enabling the extraction of structure-informed embeddings trained via SaProt. The model checkpoints for PTM-SaProt are publicly available at Google Drive. While both SaProt and PTM-Mamba rely on structural predictions from models like AlphaFold2 and ESMFold, it is important to note that these folding models do not explicitly predict or encode PTM-specific features. The training curves (provided below) demonstrate that PTM-SaProt exhibits faster convergence compared to PTM-Mamba (ESM2-650M embeddings).

Due to computational constraints with SaProt (pre-folding protein structures), we conducted 5-fold cross-validation experiments on a newly added downstream task to predict the effects of the PTM on protein-protein interaction. The results indicate that PTM-SaProt performs comparably to PTM-Transformer (using ESM2 embeddings), slightly outperforming it in metrics such as Accuracy, Precision, Recall, and MCC. However, PTM-SaProt underperforms relative to PTM-Mamba (ESM-2 embeddings). We hypothesize that the suboptimal performance of PTM-SaProt may be attributed to the lower quality of predicted structures, which could limit its representational capacity in downstream tasks. We appreciate the suggestion and have included these analyses to evaluate the generalizability of our proposed method.

“Lastly, I cannot find the details of the baseline and hyperparameter settings used for comparisons in the paper. Hyperparameter settings are crucial in machine learning algorithms, and it would be beneficial for the authors to provide an explanation of their chosen settings and present the results accordingly. We need fair comparison when claiming a methodology works.”

We detailed the baseline hyperparameters and network architectures in supplementary tables, Methods, and pseudocode (Supplementary Listings). Additionally, we release the train, validation, and test dataset of the four benchmarks at Google Drive.

“While the experiments in this paper validate PTM-Mamba from several perspectives, the depth and biological insights seem insufficient for an Article paper in Nature Methods. I would suggest that the paper be published as a brief communication, which often emphasizes its broad interest and impact. Future research can continue to explore and validate PTM-Mamba based on this foundation, and it is highly anticipated that PTM-Mamba will assist biologists in their work.”

We agree with this and have communicated this to the editor. We have also edited the entire manuscript to conform to a Brief Communications format.

NMETH-BC5553A - Response To Reviewers

Reviewer 2:

"I've looked at the revisions and rebuttals. For the most part, the authors have done a good job of addressing the reviewer concerns. I disagree with their assessment that doing the flipping properly would be too slow (yes, it's slower than the naive flip, but it's a tiny fraction of the training time), but the overall results of the paper are strong."

We greatly appreciate Reviewer 2's time and suggestions for our manuscript, which has made it substantially better. While we don't include the flipping strategy, we will explore this option in future works.

Reviewer 3:

"Thank you for your detailed explanation and revision point by point. I have read the author's comments carefully and my key technical concerns have been addressed. Please put your new results into the manuscript. In addition, the authors should pay more attention to their presentation. The current format is not suitable for brief communication. For such a paper, it is crucial to succinctly summarize the key novelties, technical contributions, and main results. The current version is not quite consistent with the expected format and needs further improvement."

We have now included all of the necessary results and have followed all editorial guidelines for formatting. Thank you so much for the critical reviews of our work and for making the manuscript substantially improved!